

# Deep-sea sediments of the global ocean

Markus Diesing[1]

[1]Geological Survey of Norway (NGU), Postal Box 6315 Torgarden, 7491 Trondheim, Norway

*Correspondence to*: Markus Diesing (markus.diesing@ngu.no)

**Abstract.** Although the deep-sea floor accounts for more than 70 % of the Earth's surface, there has been little progress in relation to deriving maps of seafloor sediment distribution based on transparent, repeatable and automated methods such as machine learning. A new digital map of the spatial distribution of seafloor lithologies in the deep sea below 500 m water depth is presented to address this shortcoming. The lithology map is accompanied by estimates of the probability of the most probable class, which may be interpreted as a spatially-explicit measure of confidence in the predictions, and probabilities

for the occurrence of seven lithology classes (Calcareous sediment, Clay, Diatom ooze, Lithogenous sediment, Mixed calcareous-siliceous ooze, Radiolarian ooze and Siliceous mud). These map products were derived by the application of the Random Forest machine learning algorithm to a homogenised dataset of seafloor lithology samples and global environmental predictor variables that were selected based on the current understanding of the controls on the spatial distribution of deep-sea sediments. The overall accuracy of the lithology map is 69.5 %, with 95 % confidence limits of 67.9 % and 71.1 %. It is

expected that the map products are useful for various purposes including, but not limited to, teaching, management, spatial planning, design of marine protected areas and as input for global spatial predictions of marine species distributions and seafloor sediment properties. The map products are available at https://doi.pangaea.de/10.1594/PANGAEA.911692 (Diesing, 2020).

**1 Introduction**

The deep-sea floor accounts for >90 % of seafloor area (Harris et al., 2014) and >70 % of the Earth's surface. It acts as a receptor of the particle flux from the surface layers of the global ocean, is a place of biogeochemical cycling (Snelgrove et al., 2018), records environmental and climate conditions through time and provides habitat for benthic organisms (Danovaro et al., 2014). Being able to map the spatial patterns of deep-sea sediments is therefore a major prerequisite for many studies

addressing aspects of marine biogeochemistry, deep-sea ecology and palaeo-environmental reconstructions.

Until recently, maps of global deep-sea sediments were essentially variants of a hand-drawn map presented by Berger (1974) and typically depicted five to six sediment types, namely calcareous ooze, siliceous ooze (sometimes split into diatom ooze and radiolarian ooze), deep-sea (abyssal) clay, terrigenous sediment and glacial sediment. Since then, Dutkiewicz et al. (2015) collated and homogenised approximately 14,500 samples from original cruise reports and interpolated them using a

support vector machine algorithm (Cortes and Vapnik, 1995). Their map displayed the spatial distribution of 13 lithologies
across the world ocean and exhibited some marked differences from earlier maps.

The controls on the distribution of deep-sea sediments have long been discussed (e.g. Seibold and Berger, 1996): Biogenous
oozes (>30 % microscopic skeletal material by weight) dominate on the deep-sea floor and their composition is controlled by
productivity in overlying surface ocean waters, dissolution during sinking and sedimentation and dilution with other

materials. The ocean is undersaturated with silica. Preservation of siliceous shells is therefore a function of shell thickness,
sinking time (water depth) and water temperature, as siliceous shells dissolve slower in colder water. The dissolution of
calcareous shells is increased with increasing pressure (water depth) and $CO_2$ content of the water (decreasing temperature).
The water depth at which the rate of supply with calcium carbonate to the sea floor equals the rate of dissolution (calcite
compensation depth; CCD) varies across ocean basins. Deep-sea clays dominate in the deepest parts of ocean basins below

the CCD. Deposition of terrigenous material is thought to be a function of proximity to land (distance to shore).

Dutkiewicz et al. (2016) investigated the bathymetric and oceanographic controls on the distribution of deep-sea sediments
with a quantitative machine-learning approach. The influence of temperature, salinity, dissolved oxygen, productivity,
nitrate, phosphate, silicate at the sea surface and bathymetry on lithogenous sediment, clay, calcareous sediment, radiolarian
ooze and diatom ooze was quantified. They found that bathymetry, sea surface temperature and sea surface salinity had the

largest control on the distribution of deep-sea sediments. Calcareous and siliceous oozes were not linked to high surface
productivity according to their analysis. Diatom and radiolarian oozes were associated with low sea surface salinities and
discrete sea surface temperature ranges.

The aim of this study is to derive a map of deep-sea sediments of the global ocean by utilising environmental predictor
variables for the development and application of a machine-learning spatial prediction model. Besides a categorical map

giving the spatial representation of seafloor types in the deep sea, probability surfaces for individual sediment classes and a
map displaying the probability of the most probable class in the final prediction will also be provided.

## 2 Data

### 2.1 Predictor variables

The initial choice of the predictor variables was informed by the current understanding of the controls on the distribution of

deep-sea sediments and the availability of data with full coverage of the deep sea at a reasonable resolution. We chose
predictor variables mentioned above, but also included sea-surface iron concentrations, which were not available to
Dutkiewicz et al. (2016), but is an important nutrient for phytoplankton (Table 1). The predictor variable raster layers from
Bio-ORACLE (Assis et al., 2018; Tyberghein et al., 2012) and MARSPEC (Sbrocco and Barber, 2013) were utilised.
Whenever available, statistics of the variable other than mean were downloaded. These included the minimum, maximum

and the range (maximum – minimum).



## 2.2 Response variable

The response variable is seafloor lithology, a qualitative multinomial variable. The seafloor sediment sample data (seafloor_data.npz) from Dutkiewicz et al. (2015) were downloaded from [ftp://ftp.earthbyte.org/papers/](ftp://ftp.earthbyte.org/papers/) [Dutkiewicz_etal_seafloor_lithology/iPython_notebook_and_input_data/](Dutkiewicz_etal_seafloor_lithology/iPython_notebook_and_input_data/). The original dataset consisted of 13 seafloor

lithology classes, while Dutkiewicz et al. (2016) simplified these to five major classes. As a compromise, seven lithology classes (Table 2) were defined: These include those five classes defined in Dutkiewicz et al. (2016), supplemented with mixed lithologies (calcareous-siliceous ooze and siliceous mud). For a detailed description of the original lithology classes refer to GSA Data Repository 2015271 ([https://www.geosociety.org/datarepository/2015/2015271.pdf](https://www.geosociety.org/datarepository/2015/2015271.pdf)).

## 3 Methods

The general workflow for building a predictive spatial model was outlined by Guisan and Zimmermann (2000). This involves five main steps: (1) Development of a conceptual model, (2) statistical formulation of the predictive model, (3) calibration (training) of the model, (4) model predictions and (5) evaluation of the model results (accuracy assessment). The conceptual model was already presented in the introduction. The remaining steps are described in the following sections. The analysis was performed in R 3.6.1 (R Core Team, 2018) and RStudio 1.2.1335 and is documented as an Executable Research

Compendium (ERC), which can be accessed at [https://o2r.uni-](https://o2r.uni-muenster.de/#/erc/GWME2voTDb5oeaQFuTWMCEMveKS1MiXm) [muenster.de/#/erc/GWME2voTDb5oeaQFuTWMCEMveKS1MiXm](https://o2r.uni-muenster.de/#/erc/GWME2voTDb5oeaQFuTWMCEMveKS1MiXm).

## 3.1 Data pre-processing

The raster layers (predictor variables) were stacked, limited to water depths below 500 m and projected to Wagner IV global equal-area projection with a pixel resolution of 10 km by 10 km.

The sample data (response variable) were pre-processed in the following way: Only samples deeper than 500 m were used and a minimum distance between sample locations of 14.5 km (the diagonal of a 10 km pixel) was enforced to limit the number of samples per pixel to one. This had also the effect of removing duplicates from the original sample dataset and reduced the number of records from 14,400 to 10,190. The data were projected to Wagner IV. Locations of the sample locations and their respective lithology class are shown in Fig. 1.

Predictor variable values were extracted for every sample location. The resulting data frame was subsequently split into a training and a test set with a ratio of 2:1 using a stratified random approach. The stratification was based on the lithology class. This ensured that the test set contained approximately the same class frequencies as the training set (Table 2).

## 3.2 Predictor variable selection

Variable selection reduces the number of predictor variables to a subset that is relevant to the problem. The aims of variable

selection are three-fold: (1) to improve the prediction performance, (2) to enable faster predictions and (3) to increase the



interpretability of the model (Guyon and Elisseeff, 2003). Predictor variables were selected in a two-step approach: Initially, the Boruta variable selection wrapper algorithm (Kursa and Rudnicki, 2010) was employed to identify all potentially important predictor variables. Wrapper algorithms identify relevant features by performing multiple runs of predictive models, testing the performance of different subsets (Guyon and Elisseeff, 2003). The Boruta algorithm creates so-called

shadow variables by copying and randomising predictor variables. Variable importance scores for predictor and shadow variables are subsequently computed with the Random Forest algorithm (see below). The maximum importance score among the shadow variables (MZSA) is determined and for every predictor variable, a two-sided test of equality is performed with the MZSA. Predictor variables that have a variable importance score significantly higher than the MZSA are deemed important, while those with a variable importance score significantly lower than the MZSA are deemed unimportant.

Tentative variables have a variable importance score that is not significantly different from the MZSA. Increasing the maximum number of iterations (maxRuns) might resolve tentative variables (Kursa and Rudnicki, 2010). Only important variables were retained for further analysis. In a subsequent step, the Boruta importance score was used to rank the remaining predictor variables. Beginning with the most important variable, correlated variables ($|r| > 0.5$) with lower importance were subsequently removed. As a result, only important and uncorrelated predictor variables are retained.

**3.3 Random Forest classification model**

The Random Forest (RF) prediction algorithm (Breiman, 2001) was chosen for the analysis due to its high predictive performance in a number of domains (Che Hasan et al., 2012; Cutler et al., 2007; Diesing et al., 2017; Diesing and Thorsnes, 2018; Huang et al., 2014; Prasad et al., 2006). The RF is an ensemble technique based on classification trees (Breiman, 1984). Randomness is introduced in two ways: by constructing each tree from a bootstrapped sample of the training data,

and by using a random subset of the predictor variables at each split in the tree growing process. As a result, every tree in the forest is unique. By aggregating the predictions over a large number of uncorrelated trees, prediction variance is reduced and accuracy improved (James et al., 2013, p. 316). The 'votes' for a specific class can be interpreted as a measure of probability for that class occurring in a specific location. The final prediction is determined by the class with the highest probability (vote count) to occur in a specific location. The randomForest package (Liaw and Wiener, 2002) was used to perform the

analysis.

RF generally performs well with default settings, i.e. without the tuning of parameters. However, it might be advisable to tune those parameters that have the largest impact on predictive accuracy. These are the number of trees in the forest ($n_{tree}$) and the number of variables to consider at any given split ($m_{try}$). Those two parameters were tuned in a grid search with the package e1071 (Meyer et al., 2019) using a 10-fold cross-validation scheme with three repeats on the training dataset.

RF also provides a relative estimate of predictor variable importance. This is measured as the mean decrease in accuracy associated with each variable when it is assigned random but realistic values and the rest of the variables are left unchanged. The worse a model performs when a predictor is randomised, the more important that predictor is in predicting the response variable. The mean decrease in accuracy was left unscaled and is reported as a fraction ranging from 0 to 1.





### 3.4 Accuracy assessment

The accuracy of the model was assessed based on a confusion matrix that was derived by predicting the model on the test set. Overall accuracy was used to evaluate the global accuracy of the model, while error of omission and error of commission were selected as class-specific metrics of accuracy. The overall accuracy gives the percentage of cases correctly allocated and is calculated by dividing the total number of correct allocations by the total number of samples (Congalton, 1991). The error of omission is the number of incorrectly classified samples of one class divided by the total number of reference

samples of that class. The error of commission is the number of incorrectly classified samples of one class divided by the total number of samples that were classified as that class (Story and Congalton, 1986). The overall accuracy, its 95% confidence intervals and a one-sided test to evaluate whether the overall accuracy was significantly higher than the no information rate (NIR) were calculated by applying the confusionMatrix() function of the caret package (Kuhn, 2008). The confidence interval is estimated using a binomial test. The NIR is taken to be the proportion of the most frequent class.

Errors of omission and commission are not provided by the function but can be calculated from the confusion matrix.

### 3.5 Environmental space

It is generally preferable to apply a suitable sampling design for model calibration and evaluation. This would ensure that the environmental variable space is sampled in a representative way. Various methods have been proposed to optimise sampling effort, including stratified random, generalised random tessellation stratified (Stevens Jr and Olsen, 2003) and conditioned

Latin hypercube sampling (Minasny and McBratney, 2006) among others. Guidelines for optimising sampling design for accuracy assessment have also been developed (Olofsson et al., 2014; Stehman and Foody, 2019). However, such approaches are not feasible here due to time and financial constraints. Instead, we utilised available (legacy) sampling data, which was split into training and accuracy testing data sets. It might nevertheless be prudent to assess to what extent the selected samples cover the environmental space of the predictor variables. This was achieved by creating a random

subsample (n = 10,000) of the selected environmental predictor variables and displaying the density distribution of the random subsample together with the density distribution of environmental variables based on the training data. This allows for a qualitative check to what degree the environmental space is sampled in a representative way.

### 4 Results

### 4.1 Variable selection

The Boruta algorithm was run with maxRuns = 500 iterations and a *p*-value of 0.05, leaving no variables unresolved (i.e. tentative). All 38 predictor variables initially included in the model were deemed important according to the Boruta analysis (Fig. 2). Correlation analysis reduced the number of retained predictor variables to eight. These were bathymetry (MS_bathy_5m),           distance           to           shore           (MS_biogeo5_dist_shore_5m),           sea-floor           maximum           temperature



(BO2_tempmax_bdmean), sea-surface temperature range (BO2_temprange_ss), sea-surface maximum primary productivity
(BO2_ppmax_ss), sea-surface maximum salinity (BO2_salinitymax_ss), sea-surface salinity range (BO2_salinityrange_ss) and sea-surface minimum silicate (BO2_silicatemin_ss). The strongest correlation between remaining predictor variables (Fig. 3) was found between bathymetry and sea-floor maximum temperature (r = 0.38), sea-surface maximum salinity range and sea-surface minimum silicate (r = -0.35) and bathymetry and distance to shore (r = -0.34). Maps of the selected predictor variables are shown in Fig. A1.

**4.2 Model tuning**


The RF model was tuned with $m_{try}$ ranging between 2 and 9 (the maximum number of predictors) and $n_{tree}$ assumed values between 100 and 2000 with steps of 100. The covered range was based on previous experience. The optimum parameter combination, based on the classification error, was $m_{try}$ = 2 and $n_{tree}$ = 600. The differences in classification error were however small, with a maximum difference of 0.95 % between the best and the worst performing combination (Fig. 4).

**4.3 Model accuracy**


The confusion matrix based on the test set is shown in Table 3. The overall accuracy of the model is 69.5 %, with 95 % confidence limits of 67.9 % and 71.1 %. This is significantly higher (p < 2.2e-16) than the NIR (44.8 %). The two dominant classes, Calcareous sediment and Clay, which make up approximately 78 % of the observations have the lowest error of omission with 13.0 % and 28.8 %, respectively. Calcareous sediment is most frequently mis-classified as Clay and vice
versa. Diatom ooze has the third lowest error of omission (41.1 %), despite being only the fifth frequent class. All other classes are rare and have high errors of omission (>65 %). Errors of commission are slightly higher than those of omission for the frequently occurring lithologies (Calcareous sediment, Clay and Diatom mud), while considerably lower for the rare classes (Lithogenous sediment, Mixed calcareous-siliceous sediment, Radiolarian ooze and Siliceous mud).

**4.4 Spatial distribution of deep-sea sediments**

Probability surfaces of individual sediment classes with verbal descriptions of likelihood (Mastrandrea et al., 2011) based on the estimated probabilities are displayed in Fig. 5. For any given pixel in the map, the final lithology class is that one with the highest probability. The probability of the most probable class might be interpreted as a spatially explicit measure of map confidence. The resulting maps of the spatial distribution of deep-sea sediments and their associated confidence are shown in Fig. 6. Calcareous sediment and Clay dominate throughout the Pacific, Atlantic and Indian Oceans, whereby Clay occupies
the deep basins and Calcareous sediment is found in shallower parts of the ocean basins. In the Southern Ocean, seafloor sediments are arranged in a banded pattern around Antarctica, with Siliceous mud closest to land, followed by Clay and Lithogenous sediment (Fig. A2). An outer ring of siliceous oozes, mainly Diatom ooze, dominates in the Southern Ocean. The width of this "opal belt" (Lisitzin, 1971) varies and in places, most notably south of South America, it is discontinuous. The Arctic Ocean appears to be dominated by Clay; however, confidence is generally low here, most likely due to the

absence of samples north of 75° N (Fig. 1). Overall, map confidence varies between 0.18 and 1. It is generally lower in the vicinity of class boundaries and higher in the geographic centre of a class.

The sea-floor lithology map bears a notable resemblance with previously published hand-drawn maps (e.g. Berger, 1974) in that the distribution of Calcareous sediment, Clay and Diatom ooze are very similar. Clear differences were however also noted: Most strikingly, our map does not display a band of Radiolarian ooze in the equatorial Pacific. This was also noted by

Dutkiewicz et al. (2015) and is likely related to the sample data, which do not show a predominance of Radiolarian ooze in the equatorial Pacific. Whether the band of Radiolarian ooze is under-represented in the sample data or in fact less prominent than previously depicted remains unresolved.

Based on the predicted distribution of lithology classes, Calcareous sediments cover approximately 155 million km$^2$ of seabed below 500 m water depth, equivalent to 48.1 % of the total area (Table 4). Clays are the second most frequent

lithology occupying 133 million km$^2$ (41.3 %). Diatom ooze, Siliceous mud and Lithogenous sediment account for 5.5 %, 2.8 % and 1.5 % of deep-sea floor, respectively. Mixed calcareous-siliceous ooze and Radiolarian ooze are the least-frequent lithologies covering just over 1 million km$^2$ of deep-sea floor.

**4.5 Predictor variable importance**

The three most important predictor variables were sea-surface maximum salinity, sea-floor maximum temperature and

bathymetry with mean decreases in accuracy between 10 % and 12 % (Fig. 7). These findings are similar to results from Dutkiewicz et al. (2016), who determined sea-surface salinity, sea-surface temperature and bathymetry as the most important controls on the distribution of deep-sea sediments. Sea-surface minimum silicate was of medium importance (8.6 % decrease in accuracy), while sea-surface temperature range, sea-surface maximum primary productivity, distance to shore and sea-surface salinity range were of lower importance (<5 % decrease in accuracy).

**4.6 Environmental space**

The environmental space (Fig. 8) is generally sampled adequately, although there is a tendency for an over-representation of shallower water depths (above 3,000 m) and areas closer to land (less than 500 km). Sea-floor maximum temperature, sea-surface temperature range, sea-surface maximum primary productivity and sea-surface maximum salinity are all slightly biased towards higher values. Sea-surface salinity range and sea-surface minimum silicate are the environmental variables

that are most closely represented by the samples.

**5 Limitations of the approach**

This study utilises legacy sampling data to make predictions of the spatial distribution of seafloor lithologies in the deep-sea. This is the only viable approach as it is unrealistic to finance and execute a survey programme that samples the global ocean with adequate density within a reasonable timeframe. However, this approach also has some drawbacks:





The presented spatial predictions were based on forming relationships between lithology classes and environmental predictor variables. For such a task, it would be desirable to cover the range of values of each of the predictor variables used in the model (Minasny and McBratney, 2006). Although it was not possible to design a sampling survey, it became nevertheless obvious that the environmental space is reasonably well covered, presumably because of the relatively large number of observations, which was achievable as there was virtually no cost associated with "collecting" the samples. However, it

might not always be the case that a large sample dataset leads to adequate coverage of the environmental space. In such a case, it might be desirable to draw a suitable sub-sample that approximates the distribution of the environmental variables.

Data originating from many cruises over long time periods are most likely heterogeneous, which might lead to increased uncertainty in the predictions. Sources of uncertainty might relate to sampling gear type, vintage and timing of sampling, representativeness of subsampling, analytical pre-treatment, inconsistency of classification standards and more (van Heteren

and Van Lancker, 2015). However, Dutkiewicz et al. (2015) made efforts to homogenise the data. From a total number of more than 200,000 samples, they selected 14,400 based on strict quality-control criteria. Only surface and near-surface samples that were collected using coring, drilling or grabbing methods were included. Furthermore, only samples whose descriptions could be verified using original cruise reports, cruise proceedings and core logs were retained. Their classification scheme is deliberately generalised in order to successfully depict the main types of sediments found in the

global ocean and to overcome shortcomings of inconsistent, poorly defined and obsolete classification schemes and terminologies (Dutkiewicz et al., 2015).

Additional uncertainty might be introduced through imprecise positioning of the samples, which might lead to incorrect relations between the response variable and the predictor variables. No metadata exist on the positioning accuracy or even the method of determining the position, which might give some clues on the error associated with the recorded positions.

However, the chances that this shortcoming leads to significant problems when making associations between target and predictor variables are relatively low, as the chosen model resolution of 10 km is relatively coarse when compared with positioning accuracy.

The initial choice of predictor variables was informed by the current understanding of the controls on deep-sea sedimentation (Dutkiewicz et al., 2016; Seibold and Berger, 1996). Consequently, all selected predictor variables were deemed important

(Fig. 2). The three most important predictor variables (Fig. 7) are also in good agreement with Dutkiewicz et al. (2016). However, the large errors of commission and especially omission for the rare lithologies Lithogenous sediment, Mixed calcareous-siliceous ooze, Radiolarian ooze and Siliceous mud might indicate that the environmental controls are less well represented for these sediment types. Lithogenous sediment comprises a wide range of grain-sizes (silt, sand, gravel and coarser) and proximity to land might be an insufficient predictor. In fact, distance to shore had the second lowest variable

importance.

Sedimentation rates in the deep-sea typically range on the order of 1 – 100 mm per 1000 yrs (Seibold, 1975). The sample depths in the dataset used here might have ranged from core top to a few dm. The lithologic signal might therefore be integrated over timescales of approximately 100 yrs to a few 100,000 yrs. The model hindcasts to derive the oceanographic

predictors typically cover approximately 25 yrs, while bathymetry and distance to coast might be nearly constant since global sea-level rise ceased approximately 6,700 yrs ago (Lambeck et al., 2014). Hence, there likely exists a mismatch between the time intervals, although oceanographic variables might not have changed dramatically over much longer timescales than a few decades.

## 6 Potential usage

Despite a reasonable overall map accuracy of 70 %, there is large variation in the class-specific error as well as the spatial
distribution of map confidence. It is therefore recommended to always consult the information on map confidence along with the map of seafloor lithologies.

The probability surfaces of the seven lithologies might be used as input for spatial prediction and modelling, e.g. marine species distribution modelling on a global scale, which typically lacks information on seafloor sediments, although substrate type is assumed to be an important environmental predictor. Additionally, the presented data layers might be useful for the
spatial prediction of sediment properties (e.g. carbonate and organic carbon content).

The categorical map might serve as a resource for education and teaching, provide context for research pertaining to the global seafloor, support marine planning, management and decision-making and underpin the design of marine protected areas globally. Additionally, the provided lithology map might be useful for survey planning, especially in conjunction with confidence information to target areas where a certain lithology is most likely to occur. Conversely, areas of low confidence
could be targeted to further improve the accuracy of and confidence in the global map of deep-sea sediments.

## 7 Data availability

The presented model results (probability surfaces of the seven lithologies, lithology map and associated confidence map) are archived at https://doi.pangaea.de/10.1594/PANGAEA.911692 (Diesing, 2020).

## 8 Conclusions

Based on a homogenised dataset of seafloor lithology samples (Dutkiewicz et al., 2015) and global environmental predictor variables from Bio-ORACLE (Assis et al., 2018; Tyberghein et al., 2012) and MARSPEC (Sbrocco and Barber, 2013) it was possible to spatially predict the distribution of deep-sea sediments globally. The general understanding about the controls on deep-sea sedimentation helped building a spatial model that gives a good representation of the main lithologies Calcareous sediment, Clay and Diatom ooze, which collectively cover nearly 95% of the mapped seafloor. Further improvements should
be directed towards the controls on the distribution of rarer lithologies (Lithogenous sediment, Mixed calcareous-siliceous ooze, Radiolarian ooze and Siliceous mud).



**Author contribution**

MD designed the study, developed the model code, executed the analysis and wrote the manuscript.

**Competing interests**

The author declares no competing interests.

**Acknowledgements**

Thanks to Karl Fabian (NGU) for providing valuable feedback.

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



**Table 1: Environmental predictor variables tested in this study**

| Environmental variable | Statistics | Unit | Source |
|---|---|---|---|
| **Bathymetry** | mean | m | Sbrocco and Barber (2013) |
| **Distance to shore** | mean | km | Sbrocco and Barber (2013) |
| **Sea-surface temperature** | mean, min, max, range | °C | Assis et al. (2018) |
| **Sea-surface salinity** | mean, min, max, range | PSS | Assis et al. (2018) |
| **Sea-surface dissolved oxygen** | mean, min, max, range | $mol\ m^{-3}$ | Assis et al. (2018) |
| **Sea-surface primary productivity** | mean, min, max, range | $g\ m^{-3}\ day^{-1}$ | Assis et al. (2018) |
| **Sea-surface iron concentration** | mean, min, max, range | $\mu mol\ m^{-3}$ | Assis et al. (2018) |
| **Sea-surface nitrate concentration** | mean, min, max, range | $mol\ m^{-3}$ | Assis et al. (2018) |
| **Sea-surface phosphate concentration** | mean, min, max, range | $mol\ m^{-3}$ | Assis et al. (2018) |
| **Sea-surface silicate concentration** | mean, min, max, range | $mol\ m^{-3}$ | Assis et al. (2018) |
| **Sea-floor temperature** | mean, min, max, range | °C | Assis et al. (2018) |





**Table 2: Seafloor lithology classes used in this study, their abbreviations, their relationships to classes in Dutkiewicz et al. (2015) and the number and percentage of samples in the training and test datasets. Not included are Ash and volcanic sand/gravel, Sponge spicules and Shells and coral fragments of the original classification.**


| Lithology class | Abbreviation | Relation to Dutkiewicz et al. (2015) | Training dataset | Test dataset |
|---|---|---|---|---|
| **Calcareous sediment** | Calc.Sed | Calcareous ooze<br>Fine-grained calcareous sediment | 3029 (44.8 %) | 1512 (44.8 %) |
| **Clay** | Clay | Clay | 2219 (32.8 %) | 1108 (32.8 %) |
| **Diatom ooze** | Dia.Ooze | Diatom ooze | 361 (5.3 %) | 180 (5.3 %) |
| **Lithogenous sediment** | Lith.Sed | Gravel and coarser<br>Sand<br>Silt | 438 (6.5 %) | 218 (6.4 %) |
| **Mixed calcareous-siliceous ooze** | Mxd.Ooze | Mixed calcareous-siliceous ooze | 123 (1.8 %) | 62 (1.8 %) |
| **Radiolarian ooze** | Rad.Ooze | Radiolarian ooze | 60 (0.9 %) | 30 (0.9 %) |
| **Siliceous mud** | Sil.Mud | Siliceous mud | 539 (8.0 %) | 269 (8.0 %) |





**Table 3: Confusion matrix. Observed (reference) classes are in columns, predicted classes in rows.**

|  | Calc.Sed | Clay | Dia.Ooze | Lith.Sed | Mxd.Ooze | Rad.Ooze | Sil.Mud | Row total | Error of commission |
|---|---|---|---|---|---|---|---|---|---|
| **Calc.Sed** | 1316 | 229 | 11 | 64 | 47 | 5 | 52 | 1724 | 0.237 |
| **Clay** | 136 | 789 | 41 | 71 | 5 | 11 | 87 | 1140 | 0.308 |
| **Dia.Ooze** | 17 | 27 | 106 | 19 | 3 | 7 | 17 | 196 | 0.459 |
| **Lith.Sed** | 12 | 27 | 10 | 36 | 0 | 0 | 18 | 103 | 0.650 |
| **Mxd.Ooze** | 10 | 3 | 4 | 0 | 6 | 0 | 1 | 24 | 0.750 |
| **Rad.Ooze** | 0 | 1 | 0 | 0 | 1 | 3 | 1 | 6 | 0.500 |
| **Sil.Mud** | 21 | 32 | 8 | 28 | 0 | 4 | 93 | 186 | 0.500 |
| **Column total** | 1512 | 1108 | 180 | 218 | 62 | 30 | 269 |  |  |
| **Error of omission** | 0.130 | 0.288 | 0.411 | 0.835 | 0.903 | 0.900 | 0.654 |  |  |




**Table 4: Breakdown of areal coverage by lithology types in the global ocean below 500 m water depth.**

| Lithology | Number of pixels | Area ($10^6$ km$^2$) | Area (%) |
|---|---|---|---|
| **Calcareous sediment** | 1,546,723 | 154.672 | 48.1 |
| **Clay** | 1,328,012 | 132.801 | 41.3 |
| **Diatom ooze** | 177,545 | 17.755 | 5.5 |
| **Lithogenous sediment** | 47,469 | 4.747 | 1.5 |
| **Mixed calcareous-siliceous ooze** | 12,014 | 1.201 | 0.4 |
| **Radiolarian ooze** | 10,342 | 1.034 | 0.3 |
| **Siliceous mud** | 91,028 | 9.103 | 2.8 |
| **Sum** | 3,213,133 | 321.313 | 100 |

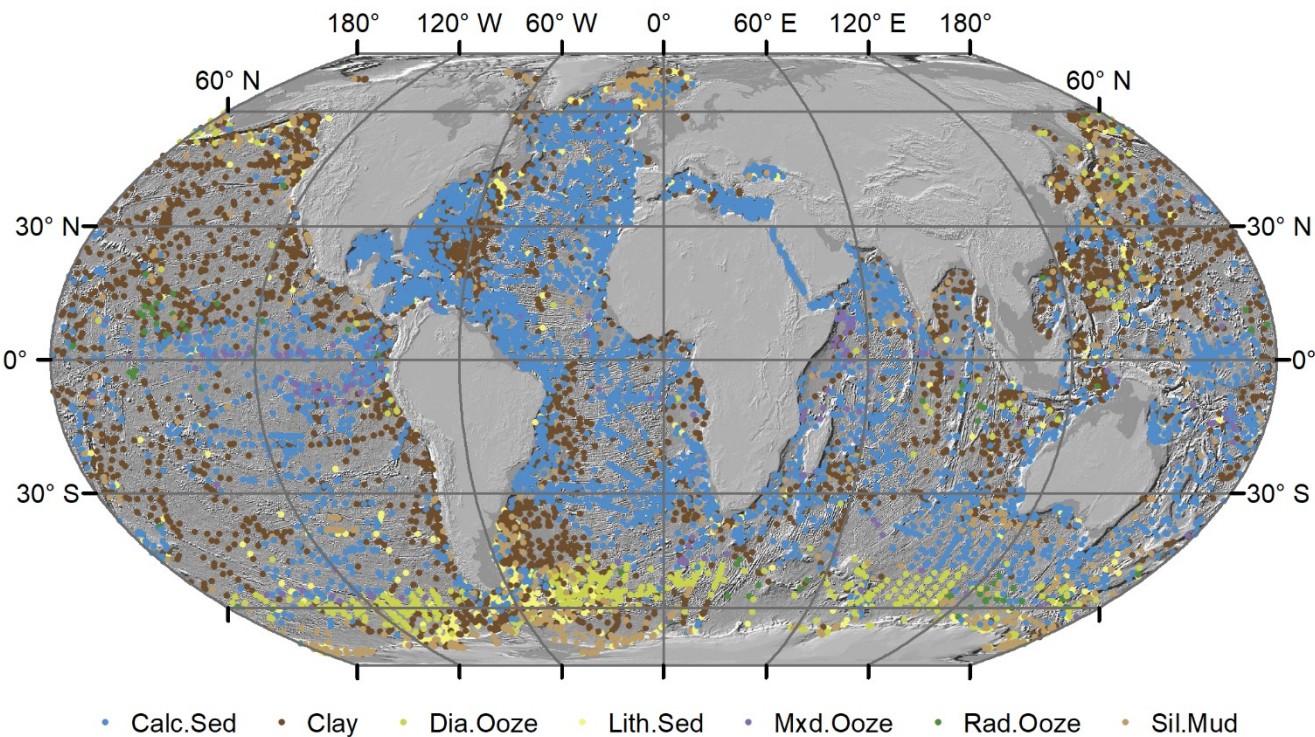

**Figure 1: Locations of samples used in this study based on data from Dutkiewicz et al. (2015). Land masses are derived from (ESRI, 2010). Hillshade topography is derived from (GEBCO, 2015).**

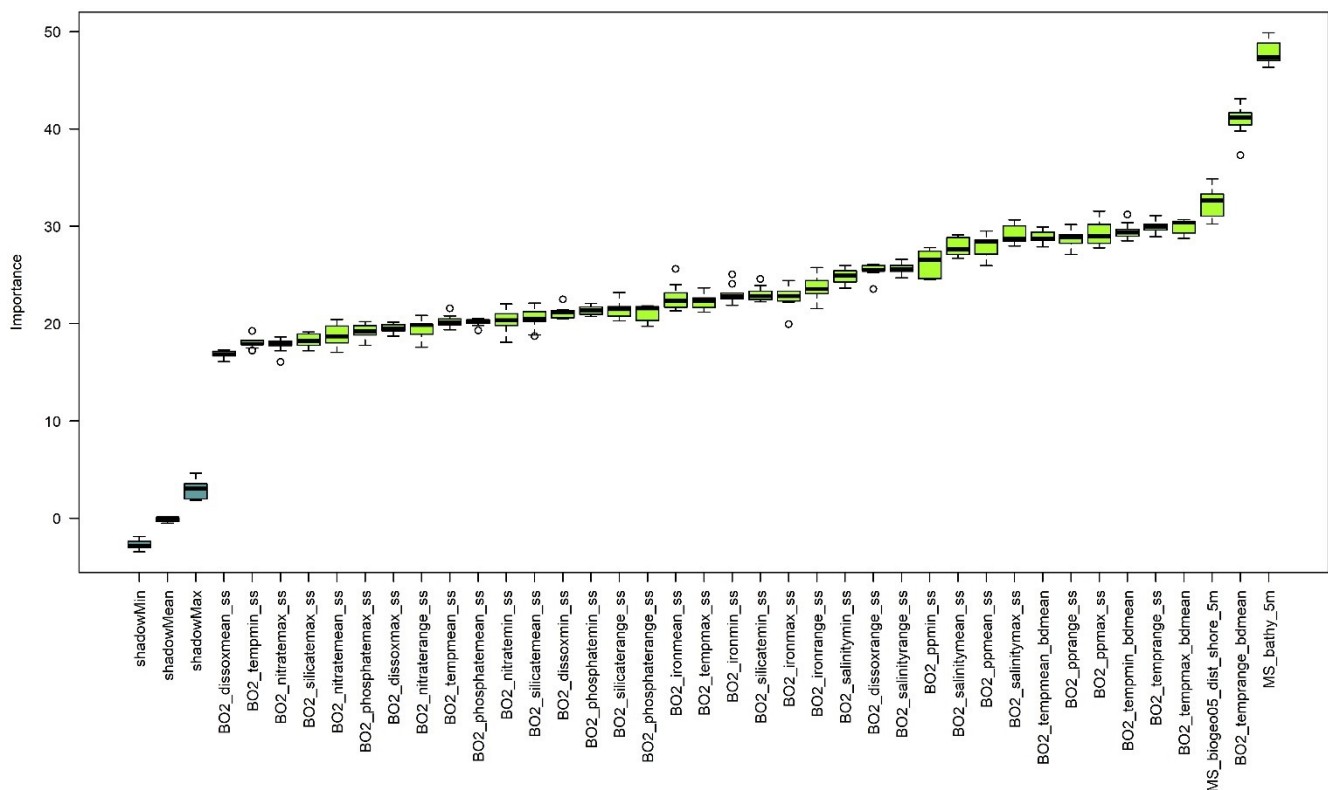

**Figure 2: Results of the Boruta variable selection process. All environmental predictor variables had an importance significantly higher than the shadow variables (shadowMin, ShadowMean and shadowMax).**






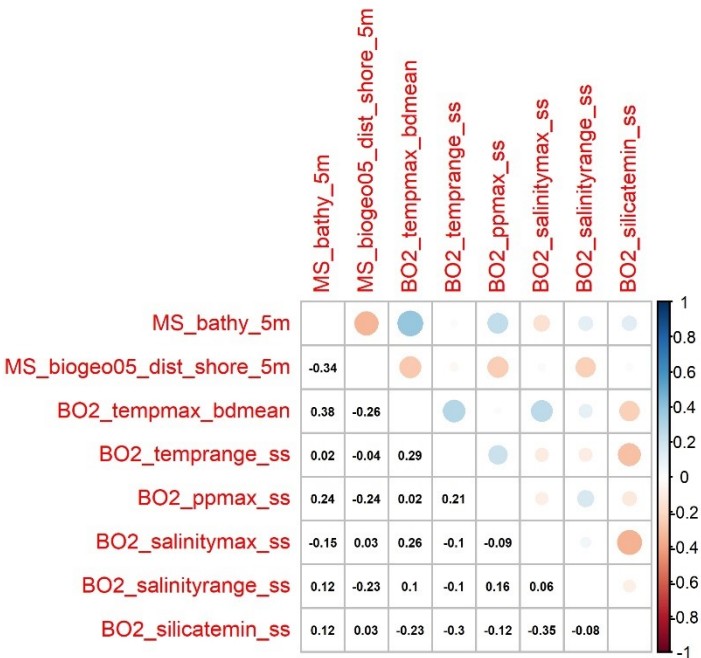

**Figure 3: Correlation plot showing the correlation coefficients of the selected predictor variables.**

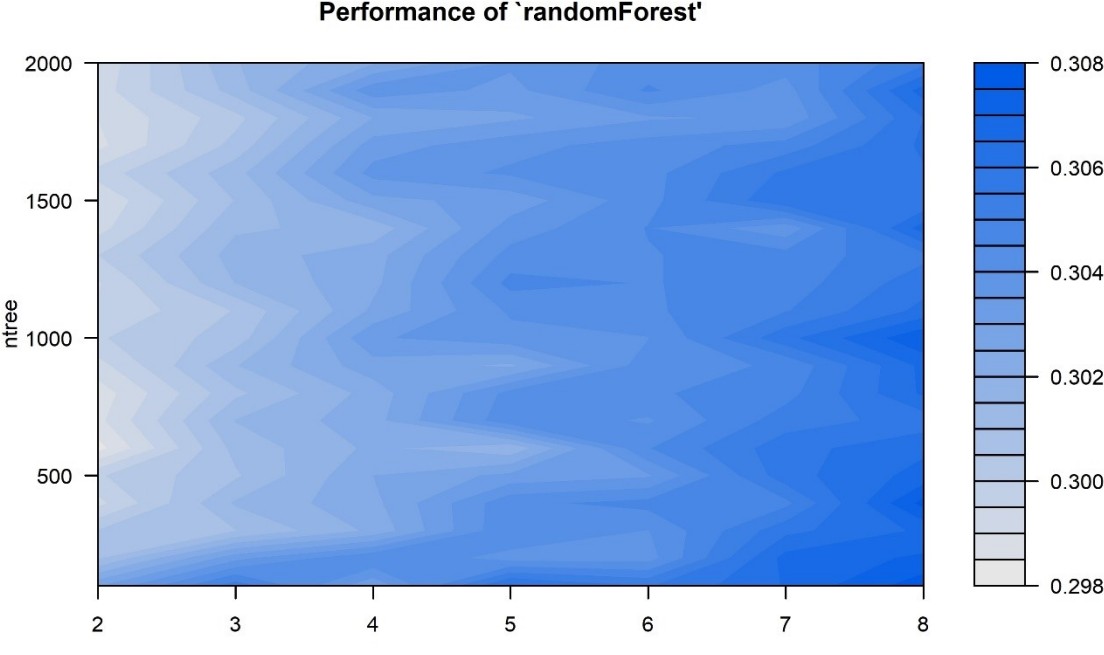

**Figure 4: Results of the model parameter tuning. The best performance (lowest error) is achieved with $n_{tree} = 600$ and $m_{try} = 2$. Differences in error are however less than 1 % between the best and worst-performing parameter combination.**




**Figure 5: Probability surfaces of the seven predicted lithologies. The verbal likelihood scale is based on Mastrandrea et al. (2011). Land masses are derived from (ESRI, 2010).**

**Figure 6: a) Predicted lithology classes and b) associated confidence in the predictions. Land masses are derived from (ESRI, 2010). Hillshade topography is derived from (GEBCO, 2015).**



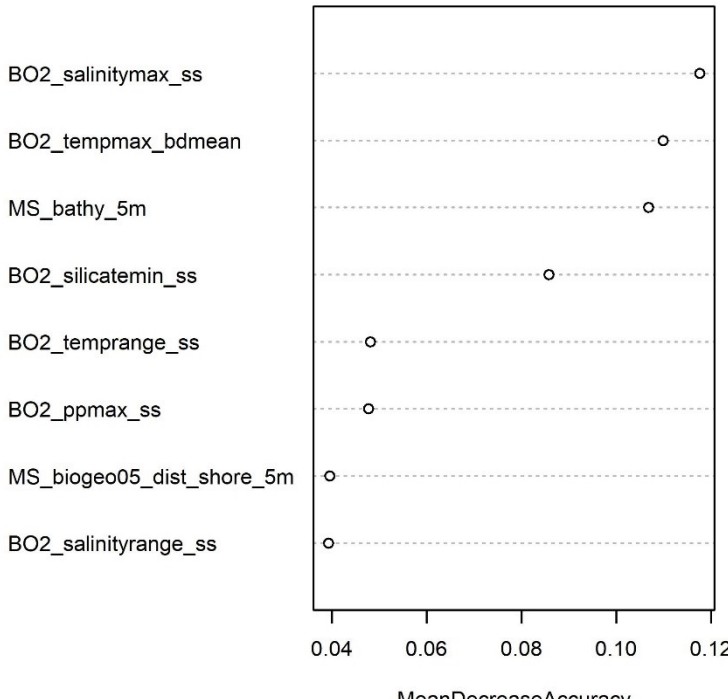

**Figure 7: Random Forest variable importance.**

**Figure 8: A visual check to what extent the samples cover the environmental space. Blue: Samples; Red: Environmental data.**



**Figure A1: Plots of the selected predictor variables. Units as in Table 1.**



**Figure A2: Predicted lithology classes in the Southern Ocean. Land masses are derived from (ESRI, 2010). Hillshade topography is derived from (GEBCO, 2015).**
