# Peer review of "Deep-sea sediments of the global ocean"

_Earth System Science Data, 2020_

## Referee Comment (RC1) · Everardo González Ávalos (Referee) · 26 May 2020

A lithology map for the seafloor below 500m depth was created with the use of the Random Forests technique. The algorithm was trained using a set of 8 global predictor variables and around 10000 measurements of seven lithology classes as response variables and achieved an accuracy of 69.5%. Two byproducts of the analysis were the probability maps for individual sediment classes and an assessment of predictor variable importances.

Since I am not an earth scientist by formation, I do not consider myself qualified to make an assessment of novelty and usefulness from a geological point of view and will not do so. I will instead comment on the general content and the technical side of the article, focusing on the materials and methods.

The article was structured clearly with a meaningful division in the following sections:
introduction, data, methods, results, limitations of the approach, potential usage, data availability, and conclusion. These sections are streamlined towards the understanding of the algorithmic implementation and its results; they retain completeness while remaining pleasantly concise, "Limitations of the approach" being the only exception to this. All accompanying figures and tables are clear and understandable, both, in digital form and in paper.

The software was tested for reproducibility using the ERC tool under https://o2r.uni-muenster.de/#/erc/GWME2voTDb5oeaQFuTWMCEMveKS1MiXm, and performed positively in this aspect. Upon closer examination, the discrepancies that led to it being flagged with failed reproducibility multiple times, appear to be minor formatting changes. The data products found under https://doi.pangaea.de/10.1594/PANGAEA.911692 are accessible, complete, and use standard file types.

For the most part, the methodology was clearly explained, with enough references to the sources for the used techniques as well as a clear specification of the software implementations used. In contrast to this, the following considerations regarding the selection of input and output variables for the Random Forest algorithm did not seem properly addressed: The importance of eliminating correlated input variables was not discussed. Considering that the Random Forests algorithm allows for non linear input relationships, a justification for this does not seem obvious, particularly since its results contradict those of the Boruta algorithm, which deemed all 38 predictors as important. The selection of 7 lithology classes as response variables instead of 5 or 13 seemed arbitrary, but the repercussions of this choice are big: the great disparity of class contribution to the test dataset (44.8% for the most represented class v.s. 0.9% for the least represented class) render model accuracy less suitable as a performance metric. The reason for this becomes evident when looking at the high errors of commission and omission of the underrepresented classes.

The above considerations do not in anyway discredit the relevance of the results, in

fact the contrary is true: the prediction accuracy for the two most common sediment classes, calcareous sediment and clay, is above the overall accuracy. In the predictions, these two classes combined make up for almost 90% of the total seabed. In contrast to this, the 0.4% share of mixed calcareous-sciuliseous ooze with an error of commission of 75% make for a statistically insignificant portion of the results; however this also makes its inclusion in this analysis questionable. The same argument can be done for the radiolarian ooze and lithogenous sediment classes.

The discussion of the predictor variable importance and the inclusion of class probability maps in the results made for a great addition to the analysis. Interpretability in machine learning is an important subject currently gaining much deserved attention and it makes an argument for the use of the Random Forests algorithm instead of other techniques such as neural networks, the latter being generally regarded as more sophisticated but requiring bigger efforts to achieve interpretability.

The overall quality of the article was satisfactory to me. It showed good understanding of the machine learning methods used and displayed outstanding transparency in their software implementation. In addition to this, the analysis of predictor variable importance and the individual probability maps in the results made for two strong points. A more detailed discussion of both, the reasoning behind the selection of predictor variable, and its effect on the results and variable predictor importance would be desirable. Since the most underrepresented response variable classes have extremely low impact on the result meaningfulness, their exclusion might be advisable. Finally, if the disparity between response variable classes should remain as large as it currently is, replacing accuracy with another performance metrics such as Intersection over Union (averaging the IoU for all individual classes) would account for a more fair result interpretation and allow for better performance comparisons in future works.

---

## Referee Comment (RC2) · Anonymous Referee #2 · 29 May 2020

General comments: The manuscript submitted by Markus Diesing presents a global scale modelling approach of the deep-sea sediments spatial distribution. This important topic for marine geology and related fields, came to the fore decades ago (the 1970s), and this paper shows how newer available global data combined with machine learning, can contribute to this direction. However, a similar approach with data from the same dataset has been presented by Dutkiewicz et al., 2015. Also, the controls on the distribution of deep-sea sediments (based on the analysis of this dataset) have been extensively discussed by Dutkiewicz et al., 2016. Both studies are mentioned in the manuscript. Nevertheless, the different applied algorithm, workflow, and the inclusion of the sea-surface iron concentration differentiate this effort. The author made a point of being very transparent about his approach, providing a thorough workflow in a well-written manuscript. Moreover, the data and code availability shows the path for similar studies promoting scientific research and knowledge.

[Figure]

Specific comments: Although the overall good quality of the manuscript, there are issues that could be addressed further. These are divided into different sections, as follows:

Manuscript structure: It is a well-constructed manuscript; however, it would be better to move the section 3.5 Environmental Space before section 3.3 Random Forest classification model. The quality characteristics (i.e. representatives) of the training samples have to be examined and assessed before any modelling approach. Similar, the results of this analysis can be presented before the model results.

Selection of predictor variables: a) The author, performed a detailed feature selection, trying to include all the relevant and not redundant predictors to this problem by applying a two-stage process: Firstly, the Boruta algorithm, and secondly a correlation filtering. While there is a detailed analysis of the Boruta algorithm, the correlation filtering is not analysed. The same is also pointed from the first Reviewer. It is not clear which correlation coefficient has been used, and the significance of these correlations. The reasons for excluding high-correlated predictor variables could be analysed further, mainly focused on the RF performance and potential bias in cases of high-correlated predictors, as in this study the variable importance & interpretation has increased value.

b) Dutkiewicz et al., 2015 and Dutkiewicz et al., 2016, referred to the absence of iron concentration as potential predictor variable form their model and correlation analysis, respectively. In this study, the author initially includes iron concentration as a potential predictor variable. However, in the final model, the iron concentration is not used. Based on the results from Boruta algorithm, the iron concentration is in a relatively high position (ranked 17th/38). Thus, further analysis of this exclusion is needed. To the same direction, it would be interesting to include a boxplot analysis between the iron concentration and the various lithology classes, enlighten further the understanding of the causal mechanisms. Such analysis is provided only for the final selected predictors inside the script.
c) The Carbonate Compensation Depth is considered as an essential factor for the distribution of the deep-sea sediments. This parameter is captured by seafloor lithology in the analysis. Nevertheless, it would be interesting to be discussed the presence of available global CCD models (and its limitations) that could be used or not as a predictor.

Random Forest (RF) modelling: The use of the RF, seems appropriate for this kind of analysis, due to interpretability that offers. The author followed an analytical workflow, including data pre-processing, stratified splitting between training & test datasets & model parameter tuning through 10-fold cross-validation. However, there are methodological issues that could be addressed further:

1a) The author uses the unscaled mean decrease in accuracy as a variable importance measure. Although this is the recommended approach (e.g. Strobl et al., 2007; Strobl et al., 2008), it would be better if the reasons behind this choice are stated inside the text.

1b) In the manuscript, it is not stated if the RF sub-sampling of predictor variables is performed with or without replacement. In the provided script seems to be with replacement (as the default option in the used package). However, studies have shown that this approach can be biased when predictor variables vary in their scale and/or in their number of categories (e.g. Strobl et al., 2007). Also, it is not also clear if any type of feature scaling has been applied in the predictor variables before modelling. In case that the author would like to continue without feature scaling for the predictor variables, packages like party seem to provide more unbiased results. In any case, it would be good to be provided with a more comprehensive explanation, as the model interpretability is one of the main targets in this work.

2a) Studies have shown that the traditional cross-validation can result in overoptimistic errors when applied in spatial data (e.g. Roberts et al., 2016). Consequently, a spatial model should also include a spatial cross-validation analysis. Moreover, in cases of

spatially unbalanced class distribution, stratified cross-validation can be applied (e.g. Lawson et al., 2017). Here, train & test samples were split with stratification, but the training was conducted only with cross-validation. Recently published spatial versions of RF could alternatively help to this direction (e.g. Hengl et al, 2018, Georganos et al, 2019). The existing dataset (despite the tremendous effort of Dutkiewicz et al., 2015) is spatially imbalanced, with areas have experienced heavier sampling efforts than other, making even more important the concept of spatial cv or/and spatial RF. The author addresses this issue by setting a minimum distance among the training points and by removing duplicates.

2b) Despite the selected class simplification, the two main classes still count for the 77.6% of the training and test sample, resulting in relatively high overall accuracy, but with limited accuracy in the rare classes. Considering the availability of methods and algorithms that try to overcome these class imbalances, (e.g. weights, penalty costs, over/under-sampling.  SMOTE, Isolation Forests) it would be interesting to see how further can be improved the performance compared to the presented (baseline) model. The overall accuracy as a performance metric is not the best option in such situations (it is also mentioned from the first Reviewer) However, the no information rate is provided, showing that there is still gain.

Results The results show good overall agreement with the above mentioned previous mapping efforts. However, the comparison demands parallel examinations of the maps from the two papers. Given the availability of the results from Dutkiewicz et al., 2015, it would be interesting to include a direct categorical map comparison between the two approaches (after the proper modifications due to the different number of lithological classes) showing clearer the areas with the highest agreement and disagreement. It is helpful that the author included the probability prediction of each class, strengthen the interpretation analysis of the results.

Technical corrections In Figures 1 & 6a, the use of purple colour for the Mixed Ooze is not ideal, as it has limited contrast with the background map and its surrounded

classes. An essential part of this study and its results is related to map creation and interpretation. Consequently, the use of colourblind-friendly palette is recommended, making the manuscript more comfortable on a broader audience.

Hope this is helpful!

Please also note the supplement to this comment:
https://www.earth-syst-sci-data-discuss.net/essd-2020-22/essd-2020-22-RC2-supplement.pdf

**Supplement:**

Review 2 on "Deep-sea sediments of the global ocean" by Markus Diesing – References

*The references are sorted based on their order inside the text, and not alphabetically or chronologically*

1. Strobl, et al, 2008: High Power! – Exploring the Statistical Properties of a Test for Random Forest Variable Importance, Proceedings of the 18th International Conference on Computational Statistics, Porto, Portugal.
2. Strobl, et al, 2008: Conditional variable importance for random forests, BMC Bioinformatics, 9, 307, https://doi.org/10.1186/1471-2105-9-307.
3. Strobl, et al, 2007.: Bias in random forest variable importance measures: Illustrations, sources and a solution, BMC Bioinformatics, 8, https://doi.org/10.1186/1471-2105-8-25.
4. Roberts, et al, 2016: Cross-validation strategies for data with temporal, spatial, hierarchical, or phylogenetic structure. Ecography. 40. 10.1111/ecog.02881.
5. Wallace et al, 2017: Decision Forests for Machine Learning Classification of Large, Noisy Seafloor Feature Sets. Computers & Geosciences. 99. 10.1016/j.cageo.2016.10.013.
6. Hengl et al, 2018:Random forest as a generic framework for predictive modeling of spatial and spatio-temporal variables. PeerJ, 6, e5518. https://doi.org/10.7717/peerj.5518
7. Georganos et al, 2019: Geographical random forests: a spatial extension of the random forest algorithm to address spatial heterogeneity in remote sensing and population modelling, Geocarto International, DOI:10.1080/10106049.2019.1595177
8. Nitesh et al, 2002: SMOTE: Synthetic Minority Over-sampling Technique. J. Artif. Intell. Res. (JAIR). 16. 321-357. 10.1613/jair.953.
9. Liu, et al, 2009: Isolation Forest. 413 – 422. 10.1109/ICDM.2008.17  Hariri, Sahand & Kind, Matias & Brunner, Robert. (2018). Extended Isolation Forest.

---

## Author Comment (AC1) · 9 Jul 2020

Referee 1 was generally positive about the submitted manuscript, but pointed out the following issues that should be addressed:

1. The importance of eliminating correlated input variables was not discussed. Considering that the Random Forests algorithm allows for non linear input relationships, a justification for this does not seem obvious, particularly since its results contradict those of the Boruta algorithm, which deemed all 38 predictors as important.

The selection of predictor variables/features is a two-step process. The first step (Boruta algorithm) narrows the selection to those variables that are relevant. In this case, all variables were identified as relevant/important. It is important to note that the Boruta algorithm is an "all-relevant" feature selection method (Nilsson et al., 2007), which identifies all predictors that might be relevant for classification (Kursa and Rudnicki, 2010). It does not address the question of redundancy in the predictor variable data. To limit redundancy, the second step seeks to identify predictor variables that are correlated with other predictors of higher importance. To do so, it is necessary to define a critical value of the correlation coefficient r. This is arguably subjective, and the choice of r will influence the number of predictor variables that are selected. To investigate the influence of r on the OOB error of a Random Forest model (using default parameter values), several values of r between 0.1 and 1 (step 0.01) were now trialled. When the OOB error is plotted over r, it is apparent that initially (with small values of r) the OOB error drops off quickly but stabilises at values of r approximately 0.4 – 0.5. As a large number of predictors potentially leads to overfitting, increases processing time and decreases interpretability of the model, it was decided to select a small value of 0.5. Referee 1 argues that the Random Forest algorithm allows for non-linear input relationships and hence the elimination of correlated input variables was unnecessary. This claim has indeed frequently been made; however, some studies recommend reducing high-dimension datasets to uncorrelated important variables (Millard and Richardson, 2015). In this case, the number of predictor variables has limited influence on prediction performance but marked influence on processing time and interpretability of the results. Therefore, a low number of predictors was selected. This selection of eight variables is also not in contradiction to the results of the Boruta analysis, as the Boruta algorithm identifies relevant features (all-relevant problem, Nilsson et al., 2007), while for model building a low number of variables that achieve comparatively high accuracy was the goal (minimal-optimal problem, Nilsson et al., 2007). I agree that the reasoning for carrying out a two-step variable selection process was not as clear as it could be. Therefore, changes were made to the relevant section in the manuscript.

2. The selection of 7 lithology classes as response variables instead of 5 or 13 seemed arbitrary, but the repercussions of this choice are big: the great disparity of class contribution to the test dataset (44.8% for the most represented class v.s. 0.9% for the least represented class) render model accuracy less suitable as a performance metric.

Following the advice from reviewer 1, it was decided to use the five classes of Dutkiewicz et al. (2016). These are broadly in line with lithology classes depicted in textbook maps and comprise Calcareous sediment, Clay, Diatom ooze, Radiolarian ooze and Lithogenous sediment. This choice still introduced large imbalances in class frequencies, as pointed out by both reviewers. To deal with the issue, a balanced version of Random Forest has now been utilised. This is achieved by using the strata and sampsize arguments of the randomForest function. In this case, we stratify by lithology class. The sampsize is then set to the same number for every class. This means that the class with the lowest frequency (Radiolarian ooze) determines the number of observations used to fit individual trees. However, each sample is still drawn from all available observations and hence this approach is likely more effective then downsampling the whole dataset prior to model building.

3. Replacing accuracy with another performance metrics such as Intersection over Union (averaging the IoU for all individual classes) would account for a more fair result interpretation and allow for better performance comparisons in future works.

I believe that overall accuracy as a metric for the performance of the model still has some meaning. Basically, if a map has an accuracy of say 60%, then 60 out of 100 randomly placed points in the map will likely be classified correctly. That information is still of importance. However, I agree that this metric is less well suited to give information on rare classes. Such information was conveiyed by the class-specific metrics error of commission and error of omission. To further address this shortcoming, I am now also providing the Balanced Error Rate (BER). The BER treats all mapped classes equally, as it is the mean error of all mapped classes, regardless of how often a class is contained in the test set. Together, both global metrics give a more comprehensive picture to assess the accuracy of the map.

References

Dutkiewicz, A., O'Callaghan, S. and Müller, R. D.: Controls on the distribution
of deep-sea sediments, Geochemistry, Geophys. Geosystems, 17(8), 3075–3098, doi:10.1002/2016GC006428, 2016.

Kursa, M. and Rudnicki, W.: Feature selection with the Boruta Package, J. Stat. Softw., 36(11), 1–11 [online] Available from: http://www.jstatsoft.org/v36/i11/paper/, 2010.

Millard, K. and Richardson, M.: On the importance of training data sample selection in random forest image classification: A case study in peatland ecosystem mapping, Remote Sens., 7(7), 8489–8515, doi:10.3390/rs70708489, 2015.

Nilsson, R., Peña, J. M., Björkegren, J. and Tegnér, J.: Consistent feature selection for pattern recognition in polynomial time, J. Mach. Learn. Res., 8, 589–612, 2007.

Please also note the supplement to this comment:
https://essd.copernicus.org/preprints/essd-2020-22/essd-2020-22-AC1-supplement.zip

―――――――――――――――――――

---

## Author Comment (AC2) · 9 Jul 2020

Referee 2 attested overall good quality, but identified several issues that could be further addressed:

1. Move the section 3.5 Environmental Space before section 3.3 Random Forest classification model.

Agreed, the section was moved accordingly.

2. Similar, the results of this analysis can be presented before the model results.

Agreed, the section was moved accordingly.

3. It is not clear which correlation coefficient has been used, and the significance of these correlations. The reasons for excluding high-correlated predictor variables could

be analysed further, mainly focused on the RF performance and potential bias in cases of high-correlated predictors.

The general strategy for predictor variable selection was to initially find all predictors that are potentially relevant, then reduce redundancy by limiting the set of predictors to those that are uncorrelated. Such an approach has also been advocated by Millard and Richardson (2015). Finding important predictors is achieved with the Boruta algorithm, an "all-relevant" feature selection method (Nilsson et al., 2007), which identifies all predictors that might be relevant for classification (Kursa and Rudnicki, 2010). Limiting redundancy is subsequently achieved by a correlation analysis. To do so, it is necessary to define a critical value of the correlation coefficient r. This is arguably subjective, and the choice of r will influence the number of predictor variables that are selected. To investigate the influence of r on the OOB error of a Random Forest model (using default parameter values), several values of r between 0.1 and 1 (step 0.01) were now trialled. When the OOB error is plotted over r, it is apparent that initially (with small values of r) the OOB error drops off quickly but stabilises at values of r approximately 0.4 – 0.5. As many predictors potentially lead to overfitting, increase processing time, and decrease interpretability of the model, it was decided to select a small value of 0.5. This led to eight predictors being selected.

For clarity, the section on predictor variable selection was updated.

4. Dutkiewicz et al., 2015 and Dutkiewicz et al., 2016, referred to the absence of iron concentration as potential predictor variable form their model and correlation analysis, respectively. In this study, the author initially includes iron concentration as a potential predictor variable. However, in the final model, the iron concentration is not used.

Dutkiewicz et al. (2016) highlighted that iron concentration in surface waters might be an important predictor, as phytoplankton blooms (e.g. diatoms) are enhanced by iron fertilisation. However, they also point out that the Southern Ocean where diatom oozes are abundant, receives very little iron. In fact, productivity is not the only factor

determining the composition of deep-sea sediments, dissolution during sinking and dilution with other materials also have to be considered.

Iron concentration was initially included in this study. However, iron concentration was not included in the final model because it was correlated (above the selected threshold) with sea-floor minimum temperature. Box plots of iron concentration versus seafloor lithology did not reveal high discriminatory power of this predictor (see attached figures). Because of that and because it was intended to keep the variable selection process as "automated" as possible, it was decided not to intervene and force the inclusion of iron concentration as a finally selected predictor.

5. It would be interesting to be discussed the presence of available global CCD models (and its limitations) that could be used or not as a predictor.

The Carbonate Compensation Depth (CCD) is potentially another important predictor that was not included in the model. This was due to the absence of relevant datasets in the utilised databases. A literature search did not yield any publications with associated datasets that could be used. However, it is assumed that this missing information is (at least partly) provided by other predictors, such as water depth and bottom water temperature.

6. The author uses the unscaled mean decrease in accuracy as a variable importance measure. Although this is the recommended approach (e.g. Strobl et al., 2007; Strobl et al., 2008), it would be better if the reasons behind this choice are stated inside the text.

Agreed. The text has been updated accordingly and now reads:

RF also provides a relative estimate of predictor variable importance. The importance() function of the randomForest package allows to assess variable importance as the mean decrease in either accuracy or node purity. However, the latter approach might be biased when predictor variables vary in their scale of measurement or their number of categories (Strobl et al., 2007) and was not used here. Variable importance is therefore measured as the mean decrease in accuracy associated with each variable when it is assigned random but realistic values and the rest of the variables are left unchanged. The worse a model performs when a predictor is randomised, the more important that predictor is in predicting the response variable. The mean decrease in accuracy was left unscaled as recommended by Strobl and Zeileis (2008), and is reported as a fraction ranging from 0 to 1.

7. In the manuscript, it is not stated if the RF sub-sampling of predictor variables is performed with or without replacement. In the provided script seems to be with replacement (as the default option in the used package). However, studies have shown that this approach can be biased when predictor variables vary in their scale and/or in their number of categories (e.g. Strobl et al., 2007). Also, it is not also clear if any type of feature scaling has been applied in the predictor variables before modelling. In case that the author would like to continue without feature scaling for the predictor variables, packages like party seem to provide more unbiased results. In any case, it would be good to be provided with a more comprehensive explanation, as the model interpretability is one of the main targets in this work.

Agreed. The predictor variables have now been scaled and the subsampling of predictor variables is performed without replacement. The manuscript was updated accordingly.

8. Studies have shown that the traditional cross-validation can result in overoptimistic errors when applied in spatial data (e.g. Roberts et al., 2016). Consequently, a spatial model should also include a spatial cross-validation analysis. Moreover, in cases of spatially unbalanced class distribution, stratified cross-validation can be applied (e.g. Lawson et al., 2017). Here, train & test samples were split with stratification, but the training was conducted only with cross-validation. Recently published spatial versions of RF could alternatively help to this direction (e.g. Hengl et al, 2018, Georganos et al, 2019). The existing dataset (despite the tremendous effort of Dutkiewicz et al., 2015) is

spatially imbalanced, with areas have experienced heavier sampling efforts than other, making even more important the concept of spatial cv or/and spatial RF. The author addresses this issue by setting a minimum distance among the training points and by removing duplicates.

Agreed. I have now implemented a spatial leave one out cross validation scheme to test the accuracy of the model in a more robust way. The spatial autocorrelation distance is determined with the spatialAutoRange function of the blockCV package. This value is utilised to determine the buffer size around observations which serve as a test point. Details can be found in a new section "3.5 Spatial cross-validation". However, this meant that model tuning would have become very complex and even more time-consuming.

As model tuning gave a very limited gain in performance, it was therefore decided to run the Random Forest with default parameter values.

As a result of this more robust estimation of map accuracy, the accuracy of the model is now lower (or rather less inflated), but still significantly larger than the no information rate.

9. Despite the selected class simplification, the two main classes still count for the 77.6% of the training and test sample, resulting in relatively high overall accuracy, but with limited accuracy in the rare classes. Considering the availability of methods and algorithms that try to overcome these class imbalances, (e.g. weights, penalty costs, over/under-sampling. SMOTE, Isolation Forests) it would be interesting to see how further can be improved the performance compared to the presented (baseline) model. The overall accuracy as a performance metric is not the best option in such situations (it is also mentioned from the first Reviewer) However, the no information rate is provided, showing that there is still gain.

The problem of class imbalances is now addressed by utilising a balanced version of Random Forest. This is achieved by using the strata and sampsize arguments of the

randomForest function. Here, we stratify by lithology class. The sampsize is then set to the same number for every class. This means that the class with the lowest frequency (Radiolarian ooze) determines the number of observations used to fit individual trees. However, each sample is still drawn from all available observations and hence this approach is likely more effective then downsampling the whole dataset prior to model building.

In addition to the overall accuracy, I have now also included the balanced error rate, which is more appropriate for datasets with unbalanced class frequencies.

As a result, the final map has a different appearance in some areas of the global ocean. It now approximates hand-drawn maps of the distribution of deep-sea sediments in much more detail. For example, equatorial patches of radiolarian ooze in the Indian Ocean are visible now. A near-equatorial band of radiolarian ooze in the eastern Pacific is now visible, too.

10. The results show good overall agreement with the above mentioned previous mapping efforts. However, the comparison demands parallel examinations of the maps from the two papers. Given the availability of the results from Dutkiewicz et al., 2015, it would be interesting to include a direct categorical map comparison between the two approaches (after the proper modifications due to the different number of lithological classes) showing clearer the areas with the highest agreement and disagreement.

It was not the intention of this contribution, and might go beyond the scope of a data description paper, to compare the final map with that of Dutkiewicz et al. (2015). I would prefer to leave such an analysis to whoever is interested in it. The map products of both publications are readily available.

11. In Figures 1 & 6a, the use of purple colour for the Mixed Ooze is not ideal, as it has limited contrast with the background map and its surrounded classes. An essential part of this study and its results is related to map creation and interpretation. Consequently, the use of colourblind-friendly palette is recommended, making the manuscript more

comfortable on a broader audience.

I agree that the choice of the colour scheme should be inclusive, but when consulting colorbrewer2.org, I could not find a colour-blind safe option for qualitative data with 5 classes. (NB: The classification has been reduced to the five classes used by Dutkiewicz et al. (2016), as suggested by reviewer 1.) The purple colour has, nevertheless, now been removed, as the respective class is no longer included. I also removed the hillshade bathymetry to make the map clearer.

References

Dutkiewicz, A., Müller, R. D., O'Callaghan, S. and Jónasson, H.: Census of seafloor sediments in the world's ocean, Geology, 43(9), 795–798, doi:10.1130/G36883.1, 2015.

Dutkiewicz, A., O'Callaghan, S. and Müller, R. D.: Controls on the distribution of deep-sea sediments, Geochemistry, Geophys. Geosystems, 17(8), 3075–3098, doi:10.1002/2016GC006428, 2016.

Kursa, M. and Rudnicki, W.: Feature selection with the Boruta Package, J. Stat. Softw., 36(11), 1–11 [online] Available from: http://www.jstatsoft.org/v36/i11/paper/, 2010.

Millard, K. and Richardson, M.: On the importance of training data sample selection in random forest image classification: A case study in peatland ecosystem mapping, Remote Sens., 7(7), 8489–8515, doi:10.3390/rs70708489, 2015.

Nilsson, R., Peña, J. M., Björkegren, J. and Tegnér, J.: Consistent feature selection for pattern recognition in polynomial time, J. Mach. Learn. Res., 8, 589–612, 2007.

Strobl, C. and Zeileis, A.: Danger: High Power! – Exploring the Statistical Properties of a Test for Random Forest Variable Importance, Munich. [online] Available from: https://epub.ub.uni-muenchen.de/2111/1/techreport.pdf, 2008.

Strobl, C., Boulesteix, A.-L., Zeileis, A. and Hothorn, T.: Bias in random forest variable

importance measures: Illustrations, sources and a solution, BMC Bioinformatics, 8(1), 25, doi:10.1186/1471-2105-8-25, 2007.

Please also note the supplement to this comment: https://essd.copernicus.org/preprints/essd-2020-22/essd-2020-22-AC2-supplement.zip

———————————————————

[Figure]

[Figure]

**Fig. 1.** Iron concentration (max)

[Figure]

**Fig. 2.** Iron concentration (mean)

[Figure]

**Fig. 3.** Iron concentration (min)

[Figure]

**Fig. 4.** Iron concentration (range)

---

## Author Response (AR1)

**General notes**

**Comments by referees in bold.**

Response in normal.

*Changes made in italics.*

All chapter, line, table, and figure numbers refer to the marked-up manuscript.

**Referee 1** was generally positive about the submitted manuscript, but pointed out the following issues that should be addressed:

**1. The importance of eliminating correlated input variables was not discussed. Considering that the Random Forests algorithm allows for non linear input relationships, a justification for this does not seem obvious, particularly since its results contradict those of the Boruta algorithm, which deemed all 38 predictors as important.**

The selection of predictor variables/features is a two-step process. The first step (Boruta algorithm) narrows the selection to those variables that are relevant. In this case, all variables were identified as relevant/important. It is important to note that the Boruta algorithm is an "all-relevant" feature selection method (Nilsson et al., 2007), which identifies all predictors that might be relevant for classification (Kursa and Rudnicki, 2010). It does not address the question of redundancy in the predictor variable data. To limit redundancy, the second step seeks to identify predictor variables that are correlated with other predictors of higher importance. To do so, it is necessary to define a critical value of the correlation coefficient r. This is arguably subjective, and the choice of r will influence the number of predictor variables that are selected. To investigate the influence of r on the OOB error of a Random Forest model (using default parameter values), several values of r between 0.1 and 1 (step 0.01) were now trialled. When the OOB error is plotted over r (Figure below), it is apparent that initially (with small values of r) the OOB error drops off quickly but stabilises at values of $r \approx 0.4 - 0.5$. As a large number of predictors potentially leads to overfitting, increases processing time and decreases interpretability of the model, it was decided to select a small value of 0.5, leading to the selection of eight predictor variables (see section 4.1 Variable selection).

[Figure]

**Influence of r-value on the out of bag error of a random forest model with default parameters. The size of the circles indicates the number of selected predictor variables (Npreds).**

Referee 1 argues that the Random Forest algorithm allows for non-linear input relationships and hence the elimination of correlated input variables was unnecessary. This claim has indeed frequently been made; however, some studies recommend reducing high-dimension datasets to uncorrelated important variables (Millard and Richardson, 2015). In this case, the number of predictor variables has limited influence on prediction performance with r-values above ≈ 0.4 but marked influence on processing time and interpretability of the results as the number of predictors increases from eight up to 38. Therefore, a low number of predictors was selected through the choice of r = 0.5. This selection of eight variables is also not in contradiction to the results of the Boruta analysis, as the Boruta algorithm identifies relevant features (all-relevant problem, Nilsson et al., 2007), while for model building a low number of variables that achieve comparatively high accuracy was the goal (minimal-optimal problem, Nilsson et al., 2007).

*Changes were made to sections 3.2 Predictor variable selection and 4.1 Variable selection (l.202-204). Figure 4 was introduced to clarify the choice of the r-value (same as figure above).*

**2. The selection of 7 lithology classes as response variables instead of 5 or 13 seemed arbitrary, but the repercussions of this choice are big: the great disparity of class contribution to the test dataset (44.8% for the most represented class v.s. 0.9% for the least represented class) render model accuracy less suitable as a performance metric.**

Following the advice from reviewer 1, it was decided to use the five classes of Dutkiewicz et al. (2016). These are broadly in line with lithology classes depicted in textbook maps and comprise Calcareous sediment, Clay, Diatom ooze, Radiolarian ooze and Lithogenous sediment.

This choice still introduced large imbalances in class frequencies, as pointed out by both reviewers. To deal with the issue, a balanced version of Random Forest has now been utilised. This is achieved by using the strata and sampsize arguments of the randomForest function. In this case, we stratify by lithology class. The sampsize is then set to the same number for every class. This means that the class with the lowest frequency (Radiolarian ooze) determines the number of observations used to fit individual trees. However, each sample is still drawn from all available observations and hence this approach is likely more effective then downsampling the whole dataset prior to model building.

*Relevant changes were made to sections 2.2 Response variable (l.65-66), 3.1 Data pre-processing (l.81-82), 3.4 Random Forest classification model (l.144-149) and Table 2.*

**3. Replacing accuracy with another performance metrics such as Intersection over Union (averaging the IoU for all individual classes) would account for a more fair result interpretation and allow for better performance comparisons in future works.**

I believe that overall accuracy as a metric for the performance of the model still has some meaning. Basically, if a map has an accuracy of say 60%, then 60 out of 100 randomly placed points in the map will likely be classified correctly. That information is still of importance. However, I agree that this metric is less well suited to give information on rare classes. Such information was conveyed by the class-specific metrics error of commission and error of omission. To further address this shortcoming, I am now also providing the Balanced Error Rate (BER). The BER treats all mapped classes equally, as it is the mean error of all mapped classes, regardless of how often a class is contained in the test set. Together, both global metrics give a more comprehensive picture to assess the accuracy of the map.

*Relevant changes were made to sections 3.6 Accuracy assessment and 4.3 Model accuracy.*

**Referee 2** attested overall good quality, but identified several issues that could be further addressed:

**1. Move the section 3.5 Environmental Space before section 3.3 Random Forest classification model.**

Agreed, the section was moved accordingly.

**2. Similar, the results of this analysis can be presented before the model results.**

Agreed, the section was moved accordingly.

**3. It is not clear which correlation coefficient has been used, and the significance of these correlations. The reasons for excluding high-correlated predictor variables could be analysed further, mainly focused on the RF performance and potential bias in cases of high-correlated predictors.**

The general strategy for predictor variable selection was to initially find all predictors that are potentially relevant, then reduce redundancy by limiting the set of predictors to those that are uncorrelated. Such an approach has also been advocated by Millard and Richardson (2015). Finding important predictors is achieved with the Boruta algorithm, an "all-relevant" feature selection method (Nilsson et al., 2007), which identifies all predictors that might be relevant for classification (Kursa and Rudnicki, 2010). Limiting redundancy is subsequently achieved by a correlation analysis. To do so, it is necessary to define a critical value of the correlation coefficient r. This is arguably subjective, and the choice of r will influence the number of predictor variables that are selected. To investigate the influence of r on the OOB error of a Random Forest model (using default parameter values), several values of r between 0.1 and 1 (step 0.01) were now trialled. When the OOB error is plotted over r, it is apparent that initially (with small values of r) the OOB error drops off quickly but stabilises at values of $r \approx 0.4 - 0.5$. As many predictors potentially lead to overfitting, increase processing time, and decrease interpretability of the model, it was decided to select a small value of 0.5. This led to eight predictors being selected.

*For clarity, section 3.2 Predictor variable selection was updated.*

**4. Dutkiewicz et al., 2015 and Dutkiewicz et al., 2016, referred to the absence of iron concentration as potential predictor variable form their model and correlation analysis, respectively. In this study, the author initially includes iron concentration as a potential predictor variable. However, in the final model, the iron concentration is not used.**

Dutkiewicz et al. (2016) highlighted that iron concentration in surface waters might be an important predictor, as phytoplankton blooms (e.g. diatoms) are enhanced by iron fertilisation. However, they also point out that the Southern Ocean where diatom oozes are abundant, receives very little iron. In fact, productivity is not the only factor determining the composition of deep-sea sediments, dissolution during sinking and dilution with other materials also have to be considered.

Iron concentration was initially included in this study. However, iron concentration was not included in the final model because it was correlated (above the selected threshold) with sea-floor minimum temperature. Box plots of iron concentration versus seafloor lithology did not reveal high discriminatory power of this predictor. Because of that and because it was intended to keep the variable selection process as "automated" as possible, it was decided not to intervene and force the inclusion of iron concentration as a finally selected predictor.

*No changes were made to the manuscript.*

**5. It would be interesting to be discussed the presence of available global CCD models (and its limitations) that could be used or not as a predictor.**

The Carbonate Compensation Depth (CCD) is potentially another important predictor that was not included in the model. This was due to the absence of relevant datasets in the utilised databases. A literature search did not yield any publications with associated datasets that could be used. However, it is assumed that this missing information is (at least partly) provided by other predictors, such as water depth and bottom water temperature.

*No changes were made to the manuscript.*

**6. The author uses the unscaled mean decrease in accuracy as a variable importance measure. Although this is the recommended approach (e.g. Strobl et al., 2007; Strobl et al., 2008), it would be better if the reasons behind this choice are stated inside the text.**

Agreed.

*The text of section 3.4 Random Forest classification model has been updated accordingly and now reads:*

*RF also provides a relative estimate of predictor variable importance. The importance() function of the randomForest package allows to assess variable importance as the mean decrease in either accuracy or node purity. However, the latter approach might be biased when predictor variables vary in their scale of measurement or their number of categories* (Strobl et al., 2007) *and was not used here. Variable importance is therefore measured as the mean decrease in accuracy associated with each variable when it is assigned random but realistic values and the rest of the variables are left unchanged. The worse a model performs when a predictor is randomised, the more important that predictor is in predicting the response variable. The mean decrease in accuracy was left unscaled as recommended by Strobl and Zeileis* (2008)*, and is reported as a fraction ranging from 0 to 1.*

**7. In the manuscript, it is not stated if the RF sub-sampling of predictor variables is performed with or without replacement. In the provided script seems to be with replacement (as the default option in the used package). However, studies have shown that this approach can be biased when predictor variables vary in their scale and/or in their number of categories (e.g. Strobl et al., 2007). Also, it is not also clear if any type of feature scaling has been applied in the predictor variables before modelling. In case that the author would like to continue without feature scaling for the predictor variables, packages like party seem to provide more unbiased results. In any case, it would be good to be provided with a more comprehensive explanation, as the model interpretability is one of the main targets in this work.**

Agreed. The predictor variables have now been scaled and the subsampling of predictor variables is performed without replacement.

*The manuscript was updated accordingly. Relevant changes were made to sections 3.1 Data pre-processing (l.80) and 3.4 Random Forest classification model (l.157-159).*

**8. Studies have shown that the traditional cross-validation can result in overoptimistic errors when applied in spatial data (e.g. Roberts et al., 2016). Consequently, a spatial model should also include a spatial cross-validation analysis. Moreover, in cases of spatially unbalanced class distribution, stratified cross-validation can be applied (e.g. Lawson et al., 2017). Here, train & test samples were split with stratification, but the training was conducted only with cross-validation. Recently published spatial versions of RF could alternatively help to this direction (e.g. Hengl et al, 2018,**

**Georganos et al, 2019). The existing dataset (despite the tremendous effort of Dutkiewicz et al., 2015) is spatially imbalanced, with areas have experienced heavier sampling efforts than other, making even more important the concept of spatial cv or/and spatial RF. The author addresses this issue by setting a minimum distance among the training points and by removing duplicates.**

Agreed. I have now implemented a spatial leave one out cross validation scheme to test the accuracy of the model in a more robust way. The spatial autocorrelation distance is determined with the spatialAutoRange function of the blockCV package. This value is utilised to determine the buffer size around observations which serve as a test point. Details can be found in a new section "3.5 Spatial cross-validation".

*A new section 3.5 Spatial cross-validation was introduced.*

However, this meant that model tuning would have become very complex and even more time-consuming. As model tuning gave a very limited gain in performance, it was therefore decided to run the Random Forest with default parameter values.

*The former sections on model tuning were deleted.*

As a result of this more robust estimation of map accuracy, the accuracy of the model is now lower (or rather less inflated), but still significantly larger than the no information rate.

*The section 4.3 Model accuracy was updated accordingly.*

**9. Despite the selected class simplification, the two main classes still count for the 77.6% of the training and test sample, resulting in relatively high overall accuracy, but with limited accuracy in the rare classes. Considering the availability of methods and algorithms that try to overcome these class imbalances, (e.g. weights, penalty costs, over/under-sampling. SMOTE, Isolation Forests) it would be interesting to see how further can be improved the performance compared to the presented (baseline) model. The overall accuracy as a performance metric is not the best option in such situations (it is also mentioned from the first Reviewer) However, the no information rate is provided, showing that there is still gain.**

The problem of class imbalances is now addressed by utilising a balanced version of Random Forest. This is achieved by using the strata and sampsize arguments of the randomForest function. Here, we stratify by lithology class. The sampsize is then set to the same number for every class. This means that the class with the lowest frequency (Radiolarian ooze) determines the number of observations used to fit individual trees. However, each sample is still drawn from all available observations and hence this approach is likely more effective then downsampling the whole dataset prior to model building.

*Relevant changes were made to section 3.4 Random Forest classification model (l.144-149).*

In addition to the overall accuracy, I have now also included the balanced error rate, which is more appropriate for datasets with unbalanced class frequencies.

*Relevant changes were made to sections 3.6 Accuracy assessment and 4.3 Model accuracy.*

As a result, the final map has a different appearance in some areas of the global ocean. It now approximates hand-drawn maps of the distribution of deep-sea sediments in much more detail. For example, equatorial patches of radiolarian ooze in the Indian Ocean are visible now. A near-equatorial band of radiolarian ooze in the eastern Pacific is now visible, too.

*Changes were made to section 4.4 Spatial distribution of deep-sea sediments, Figures 6, 7 and A2 and Table 4.*

**10. The results show good overall agreement with the above mentioned previous mapping efforts. However, the comparison demands parallel examinations of the maps from the two papers. Given the availability of the results from Dutkiewicz et al., 2015, it would be interesting to include a direct categorical map comparison between the two approaches (after the proper modifications due to the different number of lithological classes) showing clearer the areas with the highest agreement and disagreement.**

It was not the intention of this contribution, and might go beyond the scope of a data description paper, to compare the final map with that of Dutkiewicz et al. (2015). I would prefer to leave such an analysis to whoever is interested in it. The map products of both publications are readily available.

*No changes were made to the manuscript.*

**11. In Figures 1 & 6a, the use of purple colour for the Mixed Ooze is not ideal, as it has limited contrast with the background map and its surrounded classes. An essential part of this study and its results is related to map creation and interpretation. Consequently, the use of colourblind-friendly palette is recommended, making the manuscript more comfortable on a broader audience.**

I agree that the choice of the colour scheme should be inclusive, but when consulting colorbrewer2.org, I could not find a colour-blind safe option for qualitative data with 5 classes. (NB: The classification has been reduced to the five classes used by Dutkiewicz et al. (2016), as suggested by reviewer 1.) The purple colour has, nevertheless, now been removed, as the respective class is no longer included. I also removed the hillshade bathymetry to make the map clearer.

*Changes were made to Figures 1 and 6.*

**References**

[revised manuscript text omitted]

---

## Author Response (AR2)

I have requested an update of the dataset and description in the PANGAEA database; however, these changes are still awaiting implementation.

I have also requested an update of the ERC.

The input sample data are included in the PANGAEA dataset. This has now been stated explicitly in the Data Availability section of the manuscript.

Besides this, I spotted a minor error in the script: No seed was set when running the spatial leave-one-out cross validation, leading to slightly different results each time the script is run. This has now been corrected, but this has led to some minor changes in section 4.3 Model accuracy.